# Expression of 3q Oncogene SEC62 Predicts Survival in Head and Neck Squamous Cell Carcinoma Patients Treated with Primary Chemoradiation

**DOI:** 10.3390/cancers16010098

**Published:** 2023-12-24

**Authors:** Maximilian Linxweiler, Matthias Schneider, Sandrina Körner, Moritz Knebel, Lukas Alexander Brust, Felix Leon Braun, Silke Wemmert, Mathias Wagner, Markus Hecht, Bernhard Schick, Jan Philipp Kühn

**Affiliations:** 1Department of Otorhinolaryngology, Head and Neck Surgery, Saarland University Medical Center, D-66421 Homburg, Germany; matthias.schneider87@t-online.de (M.S.); sandrina.koerner@uks.eu (S.K.); moritz.knebel@uks.eu (M.K.); lukas.brust@uks.eu (L.A.B.); silke.wemmert@uks.eu (S.W.); bernhard.schick@uks.eu (B.S.); jan.kuehn@uks.eu (J.P.K.); 2Department of General and Surgical Pathology, Saarland University Medical Center, D-66421 Homburg, Germany; mathias.wagner@uks.eu; 3Department of Radiotherapy and Radiation Oncology, Saarland University Medical Center, D-66421 Homburg, Germany; markus.hecht@uks.eu

**Keywords:** head and neck squamous cell carcinoma (HNSCC), chemoradiation, SEC62, prognostic biomarkers

## Abstract

**Simple Summary:**

Head and neck cancer patients are frequently treated with primary chemoradiation, but response to therapy is hard to predict. In this study, we identified the expression of the SEC62 gene as a significant and independent predictor of patient outcome in a cohort of 127 head and neck cancer patients undergoing primary chemoradiation. Further significant prognostic factors indicating a significantly shortened overall and progression-free survival included response to therapy (RECIST1.1), lymph node metastases, distant metastases, tobacco consumption, recurrence of disease, and advanced clinical stage of disease. Together, SEC62 represents a promising and valid prognostic biomarker in this treatment setting of head and neck cancer. Its role in tumor cell biology and potential therapeutic strategies targeting SEC62 should be further investigated.

**Abstract:**

Primary chemoradiotherapy (CRT) is an established treatment option for locally advanced head and neck squamous cell carcinomas (HNSCC) usually combining intensity modified radiotherapy with concurrent platinum-based chemotherapy. Though the majority of patients can be cured with this regimen, treatment response is highly heterogeneous and can hardly be predicted. SEC62 represents a metastasis stimulating oncogene that is frequently overexpressed in various cancer entities and is associated with poor outcome. Its role in HNSCC patients undergoing CRT has not been investigated so far. A total of 127 HNSCC patients treated with primary CRT were included in this study. The median follow-up was 5.4 years. Pretherapeutic tissue samples of the primary tumors were used for immunohistochemistry targeting SEC62. SEC62 expression, clinical and histopathological parameters, as well as patient outcome, were correlated in univariate and multivariate survival analyses. High SEC62 expression correlated with a significantly shorter overall survival (*p* = 0.015) and advanced lymph node metastases (*p* = 0.024). Further significant predictors of poor overall and progression-free survival included response to therapy (RECIST1.1), nodal status, distant metastases, tobacco consumption, recurrence of disease, and UICC stage. In a multivariate Cox hazard proportional regression analysis, only SEC62 expression (*p* = 0.046) and response to therapy (*p* < 0.0001) maintained statistical significance as independent predictors of the patients’ overall survival. This study identified SEC62 as an independent prognostic biomarker in HNSCC patients treated with primary CRT. The role of SEC62 as a potential therapeutic target and its interaction with radiation-induced molecular alterations in head and neck cancer cells should further be investigated.

## 1. Introduction

Head and neck squamous cell carcinoma (HNSCC) represents the 8th most common cancer entity worldwide, accounting for more than 878,000 new cases and 444,000 deaths in 2020 [1]. Most patients are diagnosed in advanced stages of the disease, which limits therapeutic options and leads to a persistently poor five-year overall survival of approximately 60% [2]. For locally advanced (LA) HNSCC patients, primary chemoradiation is an established and frequently used therapeutic option, especially in advanced T stages and with other considerations that make surgery less preferable [2,3,4,5,6,7]. Here, intensity-modulated radiation therapy (IMRT) is combined with cisplatin, which has been shown to achieve three-year locoregional control rates of 55–80% in large-scale, prospective clinical trials [8,9,10,11,12]. However, the response to therapy is highly heterogeneous [5] with only a few established predictive biomarkers, including tumor human papillomavirus (HPV) status [13,14,15], tobacco consumption [13,15], tumor stage [13], nodal stage [13], and tumor hypoxia [16]. Hence, additional predictive and prognostic biomarkers are urgently needed for a more effective and individualized therapeutic management of LA-HNSCC patients.

The SEC62 gene, located on chromosomal region 3q26.2, encodes for an endoplasmic reticulum (ER) transmembrane protein that regulates the intracellular transport of secretory and transmembrane proteins [17,18], intracellular calcium homeostasis [19], and recovery from ER stress conditions via so called recovER-phagy [20]. Over the past several years, increasing evidence has suggested a relevant role of SEC62 in human cancer [21]. High SEC62 expression levels were associated with unfavorable prognosis in HNSCC [22], melanoma [23], breast cancer [24], and non-small cell lung cancer (NSCLC [25,26]), advanced lymph node metastasis in NSCLC [25], melanoma [23], gastric cancer [27], colorectal cancer [28], and HNSCC [22], as well as distant metastasis in breast cancer [24] and melanoma [23]. In hepatocellular carcinoma, high SEC62 expression correlated with a higher risk of tumor recurrence [29]. Recently, a pan-cancer analysis using DNA sequencing data from over 40,000 patients showed that SEC62 gene amplification represents a highly significant indicator of poor overall survival [21]. As potentially underlying molecular mechanisms, in vitro studies showed that SEC62 drives cancer metastasis by activating the MAPK/ATF2/UCA1 axis [28], mediating UPR-induced autophagy activation [27], and limiting calcium efflux across the ER membrane [19,26]. Furthermore, cancer cells benefit from high SEC62 expression levels through improved recovery from ER stress conditions which are not only induced by a high protein turnover due to elevated proliferation rate, but also by radiotherapy [30,31,32,33]. Together, these observations raise the question if SEC62-mediated ER stress tolerance also leads to an increased resistance to chemoradiotherapy (CRT). So far, SEC62 expression and its correlation with clinical parameters including therapy outcome were only investigated in HNSCC cohorts that were predominately treated with surgery [22] or in very small cohorts treated with CRT [34].

This study aimed to evaluate the influence of SEC62 expression and further clinical and histopathological parameters on treatment response and patient outcome in HNSCC patients treated with definitive chemoradiotherapy. SEC62 expression was further correlated with the patients’ clinical and histological characteristics to identify the potential effects on HNSCC biology.

## 2. Materials and Methods

### 2.1. Patients and Tissue Samples

In total, 127 HNSCC patients treated at the Department of Otorhinolaryngology, Head and Neck surgery and the Department of Radiation Oncology at the Saarland University Medical Center (Homburg, Germany) between October 2005 and August 2016 were included in this study. All the patients received primary concurrent chemoradiation (cisplatin 40 mg/m^2^ weekly or 100 mg/m^2^ three weekly + radiation with 63 to 70 Gy in 30 daily fractions). The median follow-up of the patients included in our study was 64.8 months (5.4 years). All patients gave their informed consent for their participation in our study. The study was conducted in accordance with the Declaration of Helsinki and all other relevant national ethical standards. The details on the patients’ clinical and histopathological data are shown in Table 1. The study was approved by the Saarland Medical Association ethics review board (reference number 280/10).

For immunohistochemical analyses, formalin-fixed paraffin-embedded (FFPE) tissue samples of the primary tumor that were taken during diagnostic panendoscopy prior to the start of treatment were obtained for all patients. Alcohol abuse was defined as the consumption of ≥200 g alcohol per week. Smoking behavior was categorized into four groups for further statistical analyses: lifelong non-smoker (group 1), current smoker (group 2), current reformed smoker for >15 years (group 3), and current reformed smoker for ≤15 years (group 4). The TNM and UICC stages were determined according to the 7th edition of the AJCC/UICC TNM staging system for head and neck cancers as all patients included in this study were treated before 2017.

### 2.2. Immunohistochemistry

For immunohistochemical staining targeting the proteins Sec62 and p16, FFPE tissue sections were used. First, 4 μm sections were prepared using a Leica RM 2235 microtome (Leica Microsystems, Wetzlar, Germany), transferred onto Superfrost Ultra Plus glass slides (Menzel-Gläser, Braunschweig, Germany), and dried overnight at 37 °C. After deparaffinization, the epitopes were unmasked by incubating the slides in a rice cooker with Tris-EDTA retrieval buffer (10 mM TRIS, I mM EDTA, pH 9). Afterwards, the unspecific protein binding sites were blocked by incubating the slides in PBS (phosphate-buffered saline, Sigma Aldrich, St. Louis, MO, USA) and bovine serum albumin (3% *w*/*v*; BSA, Sigma Aldrich, St. Louis, MO, USA) at pH 7.2 for 30 min at room temperature. In the next step, the slides were incubated with the primary antibody (1:1500 dilution for SEC62 and 1:4000 for p16 in 1% BSA/PBS; Anti-SEC62, clone EPR9213; ab140644, Abcam, Cambridge, UK; Anti-p16, clone 1D7D2A1; ab201980, Abcam, Cambridge, UK) for one hour at room temperature. For visualization, we used the Dako REAL detection system Alkaline Phosphatase (Dako Agilent Technologies, Glostrup, Denmark) according to the manufacturer’s instructions. Finally, the slides were counterstained with hematoxylin (Sigma Aldrich, St. Louis, MO, USA). Each immunohistochemical staining series included negative controls by omission of the primary antibody and positive controls by staining FFPE slides from a patient with an HPV-associated, SEC62 positive tonsil squamous cell carcinoma. SEC62-immunoreactivity was evaluated using the well-established immunoreactive score (IRS) by Remmele and Stegner [35] with IRS values ranging from a minimum of 0 (weak) to a maximum of 12 (strong). P16-immunoreactivity was valued as either negative or positive. All immunohistochemical stainings were valued by three experienced examiners, including one board certified histopathologist, and the mean IRS scores were finally used for statistical analyses.

### 2.3. Statistical Analysis

For statistical analyses, Prism 9 software (GraphPad Software, Boston, MA, USA) was used. To check the acquired data for Gaussian distribution, the Anderson–Darling test, D’Agostino and Pearson test, Shapiro–Wilk test, and Kolmogorov–Smirnov test were used. If the data passed ≥2 of the normality tests, parametric tests were used for statistical testing (unpaired *t* test with Welch’s correction). If the data did not pass ≥2 of the aforementioned normality tests, non-parametric tests were used (Mann–Whitney U test). Univariate survival analyses were performed using the Kaplan–Meier method and a log-rank test. For multivariate survival analyses, the Cox proportional hazard regression model was used. The fitting between the regression model and the analyzed data was evaluated using Akaike’s information criterion (AIC) and the Wald test. *p* values < 0.05 were considered statistically significant (α = 0.05) and are indicated in the figures. The tests that were used for statistical testing are indicated in the figure legends or the text, respectively.

## 3. Results

### 3.1. SEC62 Expression in HNSCC Patients and Correlation with Clinical Data as Well as Patient Outcome

First, the expression of 3q oncogene SEC62 was analyzed for all patients included in this study using immunohistochemical staining of pre-therapeutic FFPE tissue samples and quantified by an immunoreactive score (IRS, see Figure 1A,B). Thereby, 98 cases had a positive staining with a median IRS of 6.93 (2 patients with an IRS 2–3 resp. weak expression; 40 patients with an IRS 4–8 resp. moderate expression; 56 patients with an IRS 9–12 resp. strong expression) while only 29 patients were negative for SEC62 (IRS 0–1).

When correlating SEC62 expression with the patients’ overall (OS) and progression-free survival (PFS), we found a significantly longer OS (*p* = 0.0145) and a trend towards prolonged PFS in Sec62 negative compared to SEC62 positive patients (Figure 1C,E). No difference in OS and PFS was observed between patients with weak, moderate, and strong SEC62 expression (Figure 1D). Regarding a potential correlation between SEC62 expression and response to treatment according to RECIST 1.1 criteria, no significant difference in SEC62 levels was seen between patients with complete remission (CR), partial remission (PR), stable disease (SD), and progressive disease (PD), with only a slight trend towards lower SEC62 levels in CR patients compared to non-CR patients (PR + SD + PD; Figure 1F).

In the next step, we correlated SEC62 expression levels on tumor cells with the patients’ clinical and histopathological data, including tumor size, nodal status, distant metastasis, tumor differentiation, tumor localization, p16 status, nicotine consumption, patient age, and gender (Figure 2). Here, higher SEC62 expression levels correlated significantly with advanced nodal metastasis (*p* = 0.0237 for N0 vs. N3; *p* = 0.0491 for N2 vs. N3), p16 positivity (*p* = 0.0004), and chronic tobacco exposure (*p* = 0.0285 for smoking group 2 vs. 3; *p* = 0.0259 for smoking group 3 vs. 4). When comparing clinical and histopathological characteristics between SEC62 positive (n = 98) and SEC62 negative patients (n = 29), we found that only SEC62 positive patients showed distant metastases (n = 10) as well as UICC stage IV disease (n = 13). Additionally, only SEC62 positive patients showed p16 positive tumor cells in 20.4% of cases, while no SEC62 negative patient was p16 positive (see Appendix A).

### 3.2. Prognostic Factors Influencing Overall Survival of HNSCC Patients Undergoing Primary CRT

To identify further factors that have a significant influence on patient outcome in our cohort, we first performed univariate survival analyses using the Kaplan–Meier Method. Thereby, the treatment response according to RECIST 1.1 criteria (*p* < 0.0001 for CR vs. non-CR), absence of tumor recurrence (*p* = 0.0079), chronic tobacco exposure (*p* = 0.0148 for smoking group 1 + 3 vs. smoking group 2 + 4), advanced UICC stages (*p* = 0.0106 for UICC 1 + 2 vs. UICC3 + 4), negative nodal status (*p* = 0.0079), and absence of distant metastasis (*p* = 0.0029) significantly correlated with a prolonged overall survival (Figure 3).

Overall survival was not affected by p16 status (neither in the whole patient cohort (n = 127) nor in the oropharyngeal cancer only patient cohort (n = 55)), chronic alcohol consumption, tumor size, tumor localization, tumor differentiation, and patient age and gender.

### 3.3. Prognostic Factors influencing Progression-Free Survival of HNSCC Patients Undergoing Primary Crt

When correlating the patients’ clinical and histopathological data with progression-free survival (PFS), we identified the presence of distant metastases (*p* = 0.0112), chronic tobacco exposure (*p* = 0.0231), positive nodal status (*p* = 0.0018), tumor recurrence (*p* = 0.0038), poor response to therapy (*p* < 0.0001 for non-CR vs. CR), and advanced UICC stages (*p* = 0.0128 for UICC 3 + 4 vs. UICC 1 + 2) as indicators for a significantly shortened PFS (Figure 4).

PFS was not influenced by p16 status (neither in the whole HNSCC cohort nor in the oropharyngeal cancer only cohort), patient age and gender, chronic alcohol consumption, tumor size, tumor localization, and tumor differentiation.

### 3.4. Identification of Independent Prognostic Factors in the Primary CRT Cohort

In a next step, we performed multivariate statistical testing using the Cox proportional hazard regression method to delineate the independence of those prognostic factors that significantly correlated with overall survival in univariate log-rank testing (see Table 2): N stage (N+ vs. N0), M stage (M1 vs. M0), UICC stage (1 + 2 vs. 3 + 4), response to therapy according to RECIST1.1 (CR vs. non-CR), tumor recurrence (yes vs. no), SEC62 expression (positive vs. negative), and chronic tobacco exposure (group 2 + 4 vs. group 1 + 3). An AIC analysis (666.5 vs. 695.9) and Wald test (*p* < 0.0001) showed that our regression model was appropriate to describe the observed data.

Here, only treatment response (non-CR vs. CR; HR 2.723; 95% CI 1.709–4.354; *p* < 0.0001) and SEC62 expression (positive vs. negative; HR 1.79; 95% CI 1.035–3.276; *p* = 0.0462) proved to be independent prognostic factors in our study cohort, as shown in Table 2 As p16 is already known from the literature to represent a clinically established prognostic biomarker in HNSCC treated with primary CRT, we also included p16 expression (positive vs. negative) as covariate in our regression model but, again, achieved no statistically significant prognostic relevance (HR 1.044; 95% CI 0.523–1.926; *p* = 0.8964).

## 4. Discussion

In our study, we used FFPE tissue samples from n = 127 HNSCC patients treated with primary chemoradiation in order to investigate the potential relevance of 3q oncogene SEC62 as an indicator of patient outcome and response to therapy. We found that SEC62 positivity of tumor cells in pre-therapeutic samples was a significant and independent prognostic factor indicating a shorter overall survival and significantly correlated with advanced lymph node metastases, p16 positivity, and chronic tobacco exposure. The treatment response according to RECISTS1.1 proved to be the only other significant and independent predictor of overall survival in our study cohort. Further significant, but not independent, prognostic factors included nodal status, distant metastasis, UICC stage, tumor recurrence, and tobacco exposure. Together, this study underlines the high prognostic relevance of the SEC62 oncogene in HNSCC undergoing definitive chemoradiotherapy.

Regarding its role as an adverse prognostic biomarker, the results of our study are in line with previous reports that showed a correlation of high tumoral SEC62 expression with significantly shorter overall survival in HNSCC [22], melanoma [23], breast cancer [24], and NSCLC [25,26]. Comparably, a correlation between elevated SEC62 expression and the presence of lymph node metastases, as shown in our study (see Figure 2A), has also been reported in an earlier study of our group that investigated SEC62 and SOX2 expression in a cohort of 65 HNSCC patients and 29 cancer of unknown primary (CUP) patients undergoing surgery [22]. Here, higher SEC62 expression levels were observed in lymph node metastases compared with the primary tumor in HNSCC patients, as well as in lymph node metastases from CUP patients compared to lymph node metastases from HNSCC patients. Additionally, SEC62 expression showed a stepwise increase from N1 to N3 metastases. A correlation between higher SEC62 expression and advanced lymph node metastases has also been reported for NSCLC [25], melanoma [23], colorectal cancer [28], and gastric cancer [27]. Together, these data emphasize the metastasis stimulating effect of high SEC62 expression levels that has mechanistically been linked to the MAPK/ATF2/UCA1 axis [28], cellular calcium homeostasis [19], and UPR-induced autophagy activation [27]. Importantly, when comparing SEC62 positive with SEC62 negative patients in our study, we found that only SEC62 positive patients showed distant metastases, as well as UICC stage IV disease, which might have contributed to the worse outcome of the SEC62 positive group and underlines the aforementioned relevance of SEC62 for the molecular process of metastasis formation. Additionally, high SEC62 expression correlated with increased tobacco exposure as well as p16 positivity in our study cohort, which can hardly be explained from a molecular background and necessitates further hypothesis-generating investigations.

With respect to potential molecular mechanisms that can explain how elevated SEC62 expression mediates resistance to chemoradiation associated with poor prognosis, one can hypothesize that an increased ER stress tolerance mediated by SEC62 might counteract the therapeutic effects of CRT in head and neck cancer cells. Several studies in recent years reported that radiotherapy induces cellular ER stress, which represents one minor of the many molecular mechanisms how radiation exerts its therapeutic effects [30,31,32,33]. As it was shown that high SEC62 expression levels allow human cancer cells to better compensate ER stress conditions, HNSCCs might tolerate higher radiation doses once they overexpress SEC62 [26]. Additionally, it was shown that SEC62 promotes stemness and chemoresistance of human colorectal cancer through activating the Wnt/ß-catenin pathway, which represents a potential mechanism of SEC62-mediated resistance in human cancer cells to chemotherapy [36]. Regarding a potential use of SEC62 as a therapeutic target in HNSCC, Körner et al. have shown that antagonizing SEC62 function by a combined treatment with Calmodulin antagonist Trifluoperazin (TFP) and SERCA inhibitor Thapsigargin (TG) can sufficiently suppress lymphatic metastasis of HNSCC in vivo using an orthotopic xenograft head and neck cancer model [19]. In another study, a combination of TFP and TG markedly suppressed the growth of subcutaneously injected HNSCC cells in vivo [37]. The first clinical trials that investigated comparable therapeutic strategies in humans showed promising results that need to be confirmed by phase II and III trials [38]. Other potential strategies to inhibit SEC62 function in human cancer cells include the use of autophagy inhibitors such as hydroxychloroquine [27].

From a critical point of view, one has to wonder why p16 as an already established prognostic and predictive biomarker in oropharyngeal cancer did not show any significant correlation with patient outcome or with treatment response. Furthermore, after adjusting TNM and UICC stages to the 8th version of the AJCC staging system, no significant differences in outcome were seen. It would have been expected that a positive HPV status is associated with prolonged overall and progression-free survival, as well as improved treatment response, as it was shown in large-scale, prospective, randomized clinical trials [13,14]. One potential explanation for this discrepancy might be the comparatively low number of only 20 cases with p16 positive tumors, which hampers valid conclusions on the prognostic relevance of p16 status. Furthermore, it has to be mentioned that 90% (18/20) of the p16 positive HNSCC cases were diagnosed at UICC stages III and IV, in contrast to only 81% (87/107) of the p16 negative cases, which represents another potential bias. Additionally, p16 positive cases showed significantly higher SEC62 expression levels compared to p16 negative samples in our study cohort, which might have counteracted the well-known beneficial effects of HPV-associated tumor cell biology in terms of treatment response. Nonetheless, when adjusting survival analyses for UICC stages, as well as SEC62 expression as covariates, p16 did still not correlate with patient outcome; therefore, eventually we can only speculate why we found no prognostic significance of p16. Additionally, therapeutic protocols did not exactly match between all included cases with small differences regarding radiation doses (63 to 70 Gy in the primary tumor region) and cisplatin regimen (100 m^2^ every three weeks vs. 40 mg/m^2^ weekly), which might impair comparability between study subjects. However, we decided to exclude all patients that received a cumulative radiation dose of less than 60 Gy in the primary tumor region to delimitate insufficient radiation dose as a potential bias for the survival analyses. Furthermore, the data from clinical trials have shown that both of the aforementioned cisplatin regimens do not significantly differ in terms of patient outcome when combined with radiotherapy for primary or adjuvant treatment of locally advanced HNSCC [39,40]. Finally, the fact that we used pretherapeutic tumor biopsies for immunohistochemical SEC62 staining might lead to a sampling bias as we could not take into account a potential intra-tumoral heterogeneity of SEC62 expression. Nonetheless, SEC62 expression showed a very homogeneous expression pattern within tumors in previous studies of our group, including also HNSCC samples [22,41].

## 5. Conclusions

Taken together, we have shown in our study that SEC62 represents an independent prognostic biomarker in HNSCC patients treated with primary chemoradiotherapy with SEC62 overexpression correlating with a significantly shorter overall survival and advanced lymph node metastases. These results motivate further investigations focusing on the role of SEC62 as a potential therapeutic target in head and neck cancer and its interaction with radiation-induced molecular alterations in order to overcome treatment resistance to definitive CRT.

## Figures and Tables

**Figure 1 cancers-16-00098-f001:**
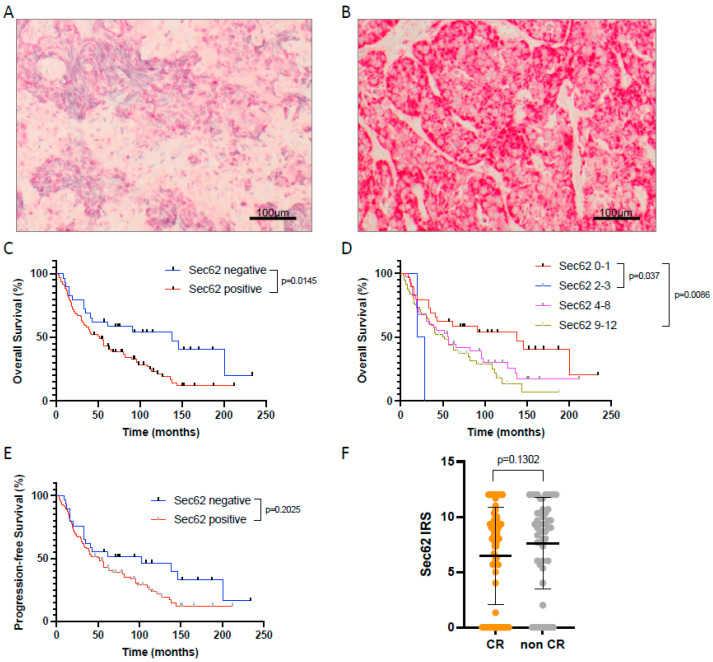
SEC62 expression in HNSCC tissue samples and impact on patient outcome. (**A**,**B**) Examples of HNSCC tissue samples with weak (**A**) and strong (**B**) SEC62 expression. (**C**) Overall survival of patients with SEC62 positive (blue) vs. SEC62 negative (red) tumors. (**D**) Overall survival of patients with no (IRS 0–1, red), weak (IRS 2–3, blue), moderate (IRS 4–8, magenta), and strong SEC62 expression (IRS 9-12, green). (**E**) Progression-free survival (PFS) of patients with SEC62 positive (red) vs. SEC62 negative tumors (blue). (**F**) Comparison of SEC62 expression in patients with complete response (CR) to CRT vs. patients without complete response (non CR; median ± standard deviation is indicated by black lines). In (**D**), only *p*-values < 0.05 are shown. IRS—immunoreactive score.

**Figure 2 cancers-16-00098-f002:**
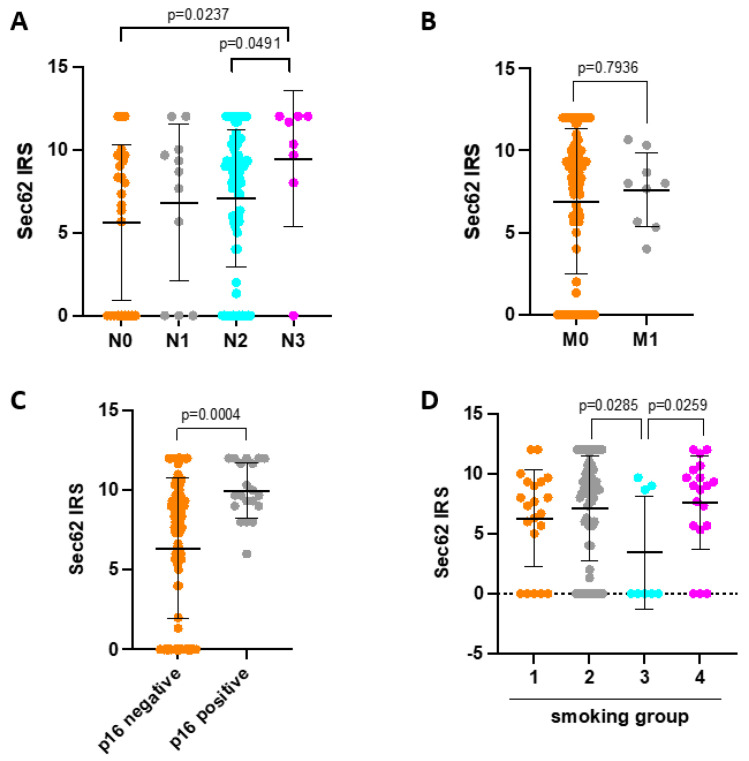
Correlation of SEC62 expression with clinical and histopathological data. (**A**) SEC62 expression in patients with N0, N1, N2, and N3 nodal status. (**B**) SEC62 expression in patients with (M1) and without (M0) distant metastasis. (**C**) SEC62 expression in patients with p16 negative vs. p16 positive tumors. (**D**) SEC62 expression depending on nicotine exposure (smoking category 1—never smoker; smoking category 2—current smoker; smoking category 3—reformed smoker ≥ 15 years; smoking category 4—reformed smoker < 15 years). In (**A**–**D**), the median ± standard deviation are indicated by black horizontal lines. In (**A**,**D**), only *p*-values < 0.05 are indicated. IRS—immunoreactive score.

**Figure 3 cancers-16-00098-f003:**
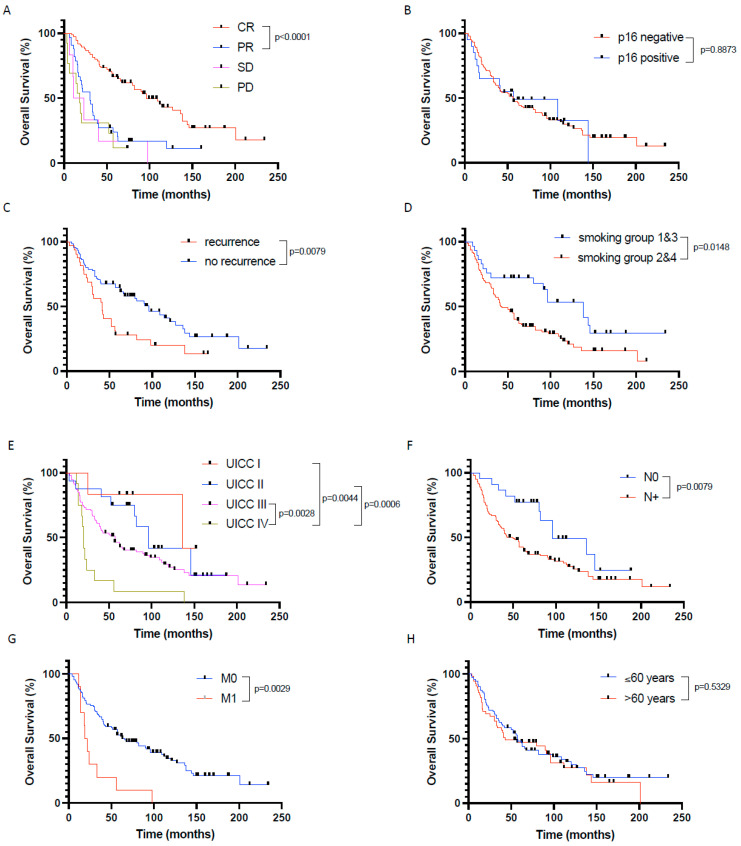
Prognostic factors significantly influencing patients’ overall survival in the investigated primary CRT cohort of HNSCC patients. Overall survival depending on (**A**) response to chemoradiation defined by RECIST1.1, (**B**) p16 expression, (**C**) recurrence vs. no recurrence, (**D**) nicotine exposure, (**E**) UICC stages, (**F**) nodal status, (**G**) distant metastasis, and (**H**) age. CR—complete remission, PR—partial remission, SD—stable disease, PD—progressive disease. In (**A**,**E**), only *p*-values < 0.05 are indicated.

**Figure 4 cancers-16-00098-f004:**
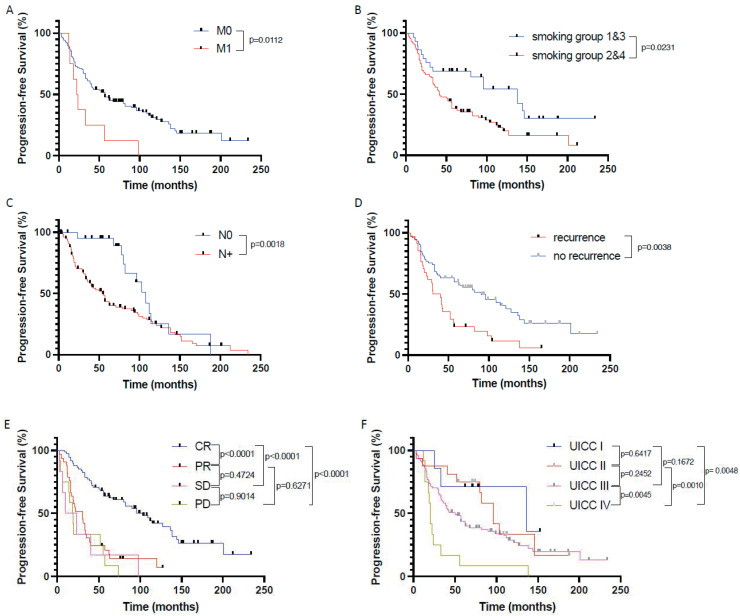
Prognostic factors significantly influencing patients’ progression-free survival in the investigated primary CRT cohort of HNSCC patients. Progression-free survival in patients (**A**) with (M1) and without (M0) distant metastasis, (**B**) depending on nicotine exposure, (**C**) depending on nodal status, (**D**) depending on tumor recurrence, (**E**) depending on response to CRT according to RECIST1.1, and (**F**) depending on UICC stages. CR—complete remission, PR—partial remission, SD—stable disease, PD—progressive disease.

**Table 1 cancers-16-00098-t001:** Clinical data of the included patients.

		Total
Number of HNSCC Patients		127
Sex	male	103 (81.1%)
	female	24 (18.9%)
age [years] ^+^		59.98
Tumor localization	Oral cavity	8 (6.3%)
	Oropharynx	55 (43.3%)
	Hypopharynx	34 (26.8%)
	Larynx	30 (23.6%)
T-Stage	T1	4 (3.2%)
	T2	21 (16.5%)
	T3	30 (23.6%)
	T4	72 (56.7%)
N-Stage	N0	25 (19.7%)
	N1	12 (9.4%)
	N2	84 (66.2%)
	N3	6 (4.7%)
M-Stage	M0	117 (92.1%)
	M1	10 (7.9%)
Grading	G1	2 (1.6%)
	G2	65 (51.1%)
	G3	60 (47.3%)
UICC-Stage	I	6 (4.7%)
	II	16 (12.6%)
	III	92 (72.4%)
	IV	13 (10.3%)
P16 expression	negative	107 (84.3%)
	positive	20 (15.7%)
Alcohol abuse	yes	41 (32.3%)
	no	74 (58.3%)
	no information	12 (9.4%)
Smoking	Lifelong non-smoker	21 (16.6%)
	Current smoker	70 (55.1%)
	Current reformed smoker for >15 years	8 (6.3%)
	Current reformed smoker for 15 ≤ years	19 (14.9%)
Response to therapy (RECIST 1.1) *	CR	75 (59.1%)
	PR	33 (26%)
	SD	6 (4.7%)
	PD	13 (10.2%)
Body Mass Index [kg/m^2^] ^+^		24.31
TSH [μIU/mL] ^+^		2.02

* CR—complete response, PR—partial response, SD—stable disease, PD—progressive disease. ^+^ average values.

**Table 2 cancers-16-00098-t002:** Multivariate analysis of prognostic factors influencing overall survival using Cox proportional hazard regression.

Variable	Hazard Ratio (HR)	95% CI	*p* Value
N Stage (N+ vs. N0)	7.279	0.2169–737.4	0.4724
M Stage (M1 vs. M0)	1.707	0.8633–3.205	0.1076
UICC Stage (1&2 vs. 3&4)	0.197	0.002–6.919	0.5558
**RECIST 1.1 (non-CR vs. CR)**	**2.723**	**1.709–4.354**	**<0.0001**
Recurrence (yes vs. no)	1.39	0.859–2.211	0.1705
**SEC62 (pos. vs. neg.)**	**1.79**	**1.035–3.276**	**0.0462**
Tobacco exposure (2&4 vs. 1&3)	1.178	0.706–2.037	0.5439

## Data Availability

The data presented in this study are available on request from the corresponding author. The data are not publicly available due to privacy restrictions.

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
