# Peer review of "Expression of 3q Oncogene SEC62 Predicts Survival in Head and Neck Squamous Cell Carcinoma Patients Treated with Primary Chemoradiation"

_cancers, 2023, doi:10.3390/cancers16010098_

Round 1

Reviewer 1 Report

Comments and Suggestions for Authors

This study investigates SEC62 gene expression in head and neck carcinoma, evaluating its role in prognosis and treatment response. This study included a relatively larger number of patients compared to previous reports in 2017; however, the patient population was heterogeneous.
#1. As mentioned by the authors in the discussion section, p16-positive and p16-negative oropharyngeal cancers did not exhibit the same prognosis. Is there any change in survival when UICC staging of cancers is updated according to the 8th AJCC edition?

#2. Within this article, I am not aware of the characteristics of patients who tested positive for SEC62, such as anatomical location (oral cavity, oropharynx...), p16 status, and TNM staging. 

#3. There may be reasons to choose chemoradiotherapy as the primary treatment tool rather than surgery, such as advanced T staging or other considerations that make surgery less preferable. 

Reviewer 2 Report

Comments and Suggestions for Authors

Comments:

1. Any data pertaining to SEC62 and Cyclin B1 within Head and Neck Tumors. especially in the introduction and discussion sections.

2. Is there any association between SEC62 and 3q duplication syndrome? especially in the introduction and discussion sections.

3. Kindly include patients' details such as BMI, lipid panel, and TSH level if available.

4. Is any association between SEC62 and p53 in head and neck tumors? especially in the introduction section.

Round 2

Reviewer 2 Report

Comments and Suggestions for Authors

No more comments